# From Lidar Measurement to Rotor Effective Wind Speed Prediction: Empirical Mode Decomposition and Gated Recurrent Unit Solution

**DOI:** 10.3390/s23239379

**Published:** 2023-11-24

**Authors:** Shuqi Shi, Zongze Liu, Xiaofei Deng, Sifan Chen, Dongran Song

**Affiliations:** 1Hunan Provincial Key Laboratory of Grids Operation and Control on Multi-Power Sources Area, Shaoyang University, Shaoyang 422000, China; 2School of Automation, Central South University, Changsha 410083, China; humble_szy@163.com; 3School of Information Technology and Management, Hunan University of Finance and Economics, Changsha 410205, China; 4Mingyang Smart Energy Group Co., Ltd., Zhongshan 528437, China

**Keywords:** wind lidar, rotor effective wind speed prediction, empirical mode decomposition, gated recurrent unit, equilibrium optimizer

## Abstract

Conventional wind speed sensors face difficulties in measuring wind speeds at multiple points, and related research on predicting rotor effective wind speed (REWS) is lacking. The utilization of a lidar device allows accurate REWS prediction, enabling advanced control technologies for wind turbines. With the lidar measurements, a data-driven prediction framework based on empirical mode decomposition (EMD) and gated recurrent unit (GRU) is proposed to predict the REWS. Thereby, the time series of lidar measurements are separated by the EMD, and the intrinsic mode functions (IMF) are obtained. The IMF sequences are categorized into high-, medium-, and low-frequency and residual groups, pass through the delay processing, and are respectively used to train four GRU networks. On this basis, the outputs of the four GRU networks are lumped via weighting factors that are optimized by an equilibrium optimizer (EO), obtaining the predicted REWS. Taking advantages of the measurement information and mechanism modeling knowledge, three EMD–GRU prediction schemes with different input combinations are presented. Finally, the proposed prediction schemes are verified and compared by detailed simulations on the BLADED model with four-beam lidar. The experimental results indicate that compared to the mechanism model, the mean absolute error corresponding to the EMD–GRU model is reduced by 49.18%, 53.43%, 52.10%, 65.95%, 48.18%, and 60.33% under six datasets, respectively. The proposed method could provide accurate REWS prediction in advanced prediction control for wind turbines.

## 1. Introduction

The rotor effective wind speed (REWS), defined as the average wind speed on the rotor surface [1], is useful for designing advanced control strategies of wind turbines [2]. REWS can be estimated by the estimation methods that have been widely studied, such as power balance estimator [3], extended-Kalman-filter-based estimator [4], Kalman-filter-based estimator [5], disturbance-accommodating control [6], unknown input observer [7], immersion and invariance estimator [8]. Through these estimation methods, an accurate estimate of REWS has been achieved. However, the estimated REWS is only a reflection of wind speed information at this moment, and the advanced control algorithms relying on this estimated value fail to solve the contradiction between the slow response of rotor rotation and the rapid change of wind speed.

For such contradiction, some researchers have proposed predictive control methods, which can greatly promote the power production [9,10] and reduce the operational cost [11] of wind turbines. Facing the difficulty in obtaining accurate previewed wind speed information with common measurements, the development of wind lidar measurement technology has promised a solution. Lidar is capable of proactively measuring wind speed within a certain range in front of the wind turbine, independent of the influence of aerodynamic shape and wake [12], previewing wind information in advance [13]. Since lidar can provide multi-point and multi-plane measurements with high accuracy, its measurement information could be used by the intelligent predictive control, improving the control performance of wind turbines. In this context, accurately predicting the REWS with lidar measurements is vital, but the relevant research is lacking.

The existing approach, mechanism modeling, generally estimates the effective wind speed on the virtual rotor surface first, and then deduces the REWS at the hub according to the Taylor frozen turbulence hypothesis [14]. To be specific, the horizontal wind speed at different heights in front of the wind turbine can be obtained through lidar measurement. Then, the REWS on the virtual rotor surface can be calculated through the geometric relationship of the horizontal wind speed of each height. This method requires a quantity of multi-beam lasers to obtain an accurate REWS. Meanwhile, the influence of distance weighting and wind evolution is hardly modeled by the mechanism-modeling method [15]. Thus, it is difficult to achieve a high-precision prediction of the REWS by the mechanism modeling method.

According to the best knowledge of the authors, data-driven methods have not fully been used in the prediction of REWS but have been well developed in wind speed prediction in the general space [16,17]. The data-driven method can realize reliable prediction through extracting mathematical relationships and nonlinear features hidden in historical data or samples [18,19]. This type of method mainly includes two parts: data extraction and prediction modeling. The former generally refers to data selection and information acquisition [20], while the latter refers to algorithm selection and extrapolation prediction [21]. Differently from general wind speed prediction, the prediction of REWS needs to establish a complex spatiotemporal mapping between inputs and output. Therein, the input information is the wind measurements at different heights in front of the wind turbine, and the output information is the REWS. This will weaken the correlation between input and output and increase the difficulty of prediction.

Motivated by the above discussions, this study attempts to build a data-driven model to predict the REWS using lidar measurement information and proposes a REWS prediction framework based on empirical mode decomposition (EMD) and gated recurrent unit (GRU) neural networks. EMD overcomes the problem of no adaptive basis function and can directly start decomposition without conducting advance analysis and research [22]. GRU solves the problem of gradient disappearance by introducing a gating mechanism. With a simple structure, GRU can process large-size datasets and is widely used in complex time series prediction [23]. Consequently, the prediction framework combined with EMD–GRU is expected to obtain high precision results of REWS.

Differently from wind speed prediction in the general space, the prediction of REWS based on lidar measurement information can use the multi-beam measurement information of lidar together with the mechanism-modeling knowledge of REWS. By combining the measurement information with the modeling knowledge, different prediction schemes could be designed. Thereby, three EMD–GRU schemes are proposed: one with mechanism modeling as the input, one with lidar data as the input, one with the combined input. Considering the influence of independent decomposition frequency from EMD, classification prediction based on GRU is carried out. To reduce the accumulation of prediction errors, the weighted aggregation based on an intelligent equilibrium optimizer (EO) is carried out. The innovations and contributions are as follows.

The novel data-driven prediction framework based on lidar measurement information is put forward to predict the REWS, enabling advanced predictive controls of wind turbines.Three prediction models based on the proposed EMD–GRU prediction framework are designed and compared based on professional BLADED software V4.8.The frequency classification and intelligent aggregation are presented to optimize the EMD–GRU models so as to reduce the prediction error and simplify the modeling complexity.

The remainder of this paper is constructed as bellow: the REWS calculation through lidar measurement is introduced in Section 2; the EMD–GRU prediction schemes with three different inputs are thoroughly described in Section 3; the results and discussions are provided in Section 4. Finally, Section 5 concludes the paper.

## 2. Lidar Measuring and REWS Calculation

The lidar for the control utilization is usually installed on the nacelle of the wind turbine. The lidar emits laser pulses into the atmosphere, receives the backscattering signals of atmospheric particles, and calculates the wind speed in line of sight at measuring point by analyzing the Doppler frequency shift of the emitted laser and the scattered laser. The lidar has a speed range of 50 m.

The REWS calculation can adopt the mechanism modeling, mainly including two aspects. For one thing, the horizontal wind speed at different heights in front of the wind turbine should be obtained through lidar measurement. For the other, the REWS of the virtual rotor surface should be calculated based on the horizontal wind speed of each height.

For the first aspect, the calculation of horizontal wind speed at different heights is shown in Figure 1. In Figure 1a, $V_{1},V_{2},V_{3},$ and $V_{4}$ respectively represent the wind speed measured by each laser beam, and the direction of wind speed is laser direction. Project $V_{1}$ and $V_{2}$ to obtain the horizontal wind speed at that height first. Figure 1b takes $V_{1}$ as an example to introduce the geometric relationship of the projection.

In Figure 1, the projection of $V_{1}$ on $V_{a}$ can be described as:(1)$$V_{a}=V_{1}\cos\left(\theta \right)$$
where θ represents the angle between the laser beam and the horizontal plane.

Similarly, the projection of $V_{2}$ on $V_{b}$ can also be described.

Thus, at the height of $V_{1}$ and $V_{2}$, the horizontal wind speed perpendicular to the rotor surface is described as:(2)$$u_{1}=\frac{{(V}_{a}+V_{b})\times \cos(\alpha )}{2}$$
where $u_{1}$ represents horizontal wind speed at that height, and α indicates the angle between $V_{a}$ and $u_{1}$.

The horizontal wind speed at the height of $V_{3}$ and $V_{4}$ can also be obtained. Due to the scanning characteristics of lidar, this method can be extended to the calculation of horizontal wind speed at various heights.

For the other aspect, to calculate the REWS at a virtual rotor surface, it can assume that there is a virtual wind turbine at the wind speed measuring point. The area of the virtual wind turbine is subdivided into multiple horizontal sections, taking 5 parts as an example. See Figure 2 for details. Figure 2a shows the sector area, and Figure 2b shows the REWS calculation.

The calculation for the sector area of the shaded part in Figure 2 is as follow. According to Figure 2b, the top ($A_{5}$) and bottom areas ($A_{1}$) of the circular area can be directly calculated by:(3)$$S=R^{2}{cos}^{-1}\left(\frac{R-h}{R}\right)-(R-h)\sqrt{2Rh-h^{2}}$$
where S, R, and h represent the sector area, the radius of the rotor, and the height of the sector area, respectively.

To calculate the area of $A_{2}$, the areas $A_{1}$ and $A_{2}$ can be seen as a whole sector area, and the area of $A_{1}$ can be subtracted.
(4)$$S_{A_{2}}=S_{A_{1},A_{2}}-S_{A_{1}}$$
where $S_{A_{i}}$ represents the area of $A_{i}$.

The area $A_{3}$ can be obtained using:(5)$$S_{A_{1}}+S_{A_{2}}+\frac{1}{2}S_{A_{3}}=\frac{1}{2}\pi R^{2}$$
(6)$$u_{eq}=\sqrt[{3}]{\sum_{i=1}^{n_{h}}\frac{A_{i}}{A}u_{i}^{3}}$$
where, $u_{eq}$ refers to the REWS; $n_{h}$ indicates the number of divided areas of the virtual rotor surface; $A_{i}$ and $u_{i}$ are the area and horizontal wind speed of the ith zone, respectively; A refers to the gross area.

The above calculation of REWS only represents the effective wind speed faced by the virtual rotor at the measuring spot. To improve the calculation accuracy, the number of lidar measuring points should be increased. Otherwise, the prediction accuracy will be affected. Moreover, the influence of wind evolution is hard to include in the mechanism modeling.

## 3. EMD–GRU Prediction Schemes

The proposed data-driven prediction framework based on EMD–GRU is shown in Figure 3, including three parts: the data processing based on EMD, the classification prediction based on GRU, and the weighted aggregation based on EO.

In the data processing phase, the EMD decomposition and delay processing are included. The input data are decomposed into intrinsic mode functions (IMF) and residual by EMD, and then time delay is processed for each decomposition part.

In the GRU predicting phase, each IMF component is divided into high-, medium-, or low-frequency groups according to its frequency characteristics. Together with the residuals, the four groups are predicted through the same GRU neural network. The GRU parameter is determined by EO.

In the aggregating phase, after optimizing the weight of each IMF and residual by EO algorithm, all the predicted values are aggregated to obtain the predicted REWS.

### 3.1. Data Processing Based on EMD

#### 3.1.1. Determination of Input Data

In order to ensure a strong correlation between the input information and output REWS, three schemes with different inputs are proposed:

Scheme 1: there is only one input, that is, the wind speed measured by four laser beams is first processed through mechanism modeling, and then the calculated REWS is taken as the input of the EMD–GRU model.

Scheme 2: the input of the EMD–GRU model to predict the REWS is the wind speed measured by four laser beams. 

Scheme 3: the wind speed measured by four laser beams and the REWS calculated by mechanism modeling are used as the input of the EMG-GRU model.

#### 3.1.2. Empirical Mode Decomposition

EMD, as a flexible method for non-stationary and nonlinear data decomposition, shows better adaptability and usability compared to traditional decomposition methods (like Wavelet analysis). Complex wind speed input sequences can be decomposed using EMD to obtain components with different characteristic scales, which are more regular than the original input sequence. Although there are still different degrees of non-stationarity among these components, the difficulty of non-stationarity for prediction is reduced. Since EMD decomposition has a high signal-to-noise ratio, it can improve the prediction accuracy of REWS.

All the raw data sequences of the input are decomposed into sub-sequences by EMD. For the original wind speed series *X*(*t*) measured by lidar, through EMD decomposition, it can be described as the following equation:(7)$$X\left(t\right)=\sum_{i=1}^{n}C_{i}\left(t\right)+R_{n}\left(t\right)\;$$
where $C_{i}\left(t\right)\;(i=1,2,\ldots ,n)$ denotes the decomposed IMF, and $R_{n}\left(t\right)$ is the residual of EMD.

#### 3.1.3. Delay Processing

Since there is a certain distance between the measuring spot of lidar and the blade rotor, the decomposed input data cannot be directly put into the GRU neural network for prediction. Thus, it is necessary to consider time shift of the wind, and the Taylor frozen turbulence hypothesis [24] is introduced to perform delay processing for each IMF. Time delay T under different average wind speeds can be calculated using:(8)$$T=\frac{x}{\bar{u}}\;$$
where x represents the distance between the lidar measurement spot and the lidar, while $\bar{u}$ represents the average wind speed.

### 3.2. Prediction Modeling Based on GRU Neural Network

#### 3.2.1. Frequency Classification Preparation

Due to insufficient sampling rate and spline interpolation, there are some frequency components in each IMF component spectrum that are independent of the target signals. If each IMF component is modeled, it will not only reduce work efficiency but also cause error accumulation and reduce prediction accuracy because of too many models.

Therefore, during frequency grouping, the sample entropy algorithm [25] is used to calculate the entropy of each sequence of IMF to represent the complexity of each sequence. Firstly, the entropy values of IMFs under different average wind speeds are calculated. Then, at each average wind speed, the maximum entropy value is taken as the reference value. Finally, according to 1/5 and 1/10 of the maximum entropy value, all IMF subsequences are divided into three groups: high-, medium-, and low-frequency.

#### 3.2.2. GRU Neural Network

GRU is a variant proposed by Greff et al. on the basis of long short-term Memory (LSTM), with simple structure and easy calculation [23].

GRU contains two gating units, namely update gate $z_{t}$ and reset gate $r_{t}$. The update gate controls the degree to which the state information $h_{t-1}$ of the previous moment is introduced into the current state through activation function σ, while the reset gate controls the degree to which the state information $h_{t-1}$ of the previous moment is introduced into the candidate set through activation function σ. The specific calculation formulas of GRU are as follows:(9)$$r_{t}=\sigma \left(W_{xr}x_{t}+W_{hr}x_{t-1}+b_{r}\right)$$
(10)$$z_{t}=\sigma \left(W_{xz}x_{t}+W_{hz}h_{t-1}+b_{z}\right)$$
(11)$${\tilde{h}}_{t}=\tan h\left[W_{x\tilde{h}}x_{t}+W_{h\tilde{h}}\left(r_{t}\cdot h_{t-1}\right)\right]$$
(12)$$h_{t}=(1-z_{t})\cdot h_{t-1}+z_{t}\cdot {\tilde{h}}_{t}$$
where $x_{t}$ and $h_{t}$ refer to input vector and output vector, respectively; ${\tilde{h}}_{t}$ refers to candidate activation vector; W and b represent the parameter matrices and vectors, respectively.

#### 3.2.3. EO Algorithm

EO is an optimization algorithm inspired by the physical phenomenon of dynamic balance of mass in control volume [26]. Compared to other optimizers, EO has higher optimization efficiency and fewer iterations.

The main steps of EO optimization are as follows:

Step 1: Initialization and function evaluation.
(13)$$C_{i}^{0}=C_{min}+{rand}_{i}\left(C_{max}-C_{min}\right)\;i=1,\ldots ,\;n$$
where $C_{i}^{0\;}$ is initial concentration; $C_{min}$ and $C_{max}$ are the lower limit and upper limit of variables to be optimized, respectively; ${rand}_{i}$ is the random vector between 0 and 1; and n is population number.

Step 2: Equilibrium pool and candidates ($C_{eq}$).
(14)$$C_{eq,pool}=\{C_{eq,1},C_{eq,2},C_{eq,3},C_{eq,4},C_{eq,ave}\}$$
where, $C_{eq,1},C_{eq,2},C_{eq,3},C_{eq,4}$, respectively, are the optimal solutions found in the current iteration and are mainly used to improve the global exploration ability; $C_{eq,ave}$ is the average value of the above four solutions and is mainly used to improve local development ability.

Step 3: Exponential term (F).
(15)$$F=\exp(-\lambda (t-t_{0}))$$
where λ is a random number between 0 and 1; time, $t_{0}$, stands for the initial time; time, t, is defined as a function of iteration.

Step 4: Generation rate (G).
(16)$$G=G_{0}e^{-k(t-t_{0})}$$
where $G_{0}$ is the initial value and k indicates a decay constant.

Step 5: Update the solution (C).
(17)$$C=G_{eq}+\left(C-C_{eq}\right)F+G(1-F)/\lambda V$$
where V is the control volume.

#### 3.2.4. GRU Prediction Based on EO

It can be learned from Figure 3 that four GRU neural networks are adopted in total. If the learning rate is set separately for each neural network, although the prediction effect can be improved, the process of adjusting the parameters will become complicated and the feasibility of the model could be reduced. Therefore, each GRU neural network sets the same parameters and uses EO to find the optimal learning rate.

The learning rate of GRU determines whether the fitness function can converge to the minimum, which is optimized by EO. The index RMSE taken as the fitness function of EO optimization is shown as Equation (20), in which ${\hat{f}}_{i}$ refers to the prediction value of REWS obtained through the GRU neural network and $f_{i}$ refers to the actual REWS. In order to obtain ${\hat{f}}_{i}$, the four component quantities (including high-, medium-, and low-frequency groups as well as the residual) under all average wind speeds are used as the input of the GRU neural network, and the first 3/4 of the input is used for training.

The optimization process of GRU is shown in Figure 4. First, according to Equation (13), particles are evaluated for their fitness function, and then Equation (14) is used to determine the equilibrium candidates and construct the equilibrium pool. If the fitness function has not yet converged or reached the number of iterations, loop Equations (14)–(17). 

### 3.3. Aggregation Computing Based on EO

#### 3.3.1. Aggregation Computing

Aggregation computing is conducted by:(18)$$\hat{f_{i}}=w_{1}f_{HF}+w_{2}f_{MF}+w_{3}f_{LF}+w_{4}f_{Res}$$
where $\hat{f_{i}}$ is REWS, and $w_{1},\;w_{2},\;w_{3},\;w_{4}$ are weights of $f_{HF}$ (high-frequency group), $f_{MF}\;$(medium-frequency group), $f_{LF}$ (low-frequency group) and $f_{Res}$ (residual), respectively.

There are always some errors when GRU predicts the components obtained through EMD decomposition. The prediction accuracy could be improved by optimizing the weight of the predicted values, which is performed using EO.

#### 3.3.2. Aggregation Weight Optimization with EO

EO is used to determine the weight coefficients of each frequency group and residual. The evaluation indicator RMSE Equation (20) is taken as the fitness, where ${\hat{f}}_{i}$ and $f_{i}$ refer to the aggregation computing value calculated from Equation (18) and the actual REWS, respectively. The minimum fitness function should be found in EO optimization.

The procedure of aggregation weight optimization with EO is shown in Figure 5. First, after function initialization Equation (13), the result of aggregation computing Equation (18) is calculated. Both the aggregation result and the actual REWS are used to calculate the EO fitness. Then, the current balance pool state is determined according to Equation (14). After updating the exponential term Equation (15) and generation rate Equation (16), recalculate the current solution Equation (17) to find the next population of the weight. In the process of optimization, if the EO fitness converges, it indicates that the optimization is effective, and vice versa.

## 4. Results and Discussions

The data in this study are simulated and obtained using BLADED software. For the three EMD–GRU schemes, case studies are carried out. The structure of this section is as follows. First, six wind speed datasets, parameter settings, and some evaluation criteria are given. Then, several experimental results are discussed. Finally, the comparation with other models is represented, and a conclusion is made.

### 4.1. Statistical Characteristics of Six Wind Speed Datasets

Six datasets of different average wind speeds with 12% turbulence are selected. The actual REWS are calculated and obtained by BLADED with four-beam lidar. Each dataset contains 29,500 observations, and the first 3/4 observations are used as the test set, while the last 1/4 as the verification set. Table 1 shows the statistical characteristics of the six datasets.

### 4.2. Parameter Setting of the Model

All models are implemented in MATLAB R2020a. Table 2 shows the parameters of GRU algorithm and EO algorithm. 

The learning rate of GRU neural network is determined by EO algorithm. The input dimension is determined by three EMD–GRU schemes. EO_1_ sets the learning rate of neural network, while EO_2_ sets the weight of weighted aggregation of prediction results of each component.

### 4.3. Evaluation Criteria

In order to evaluate the prediction performance, three evaluated criteria are used [27], including mean absolute error (MAE), root mean squared error (RMSE), and mean absolute percentage error (MAPE): (19)$$MAE=\frac{1}{n}\sum_{i=1}^{n}\left|f_{i}-{\hat{f}}_{i}\right|$$
(20)$$RSME=\sqrt{\frac{1}{n}\sum_{i=1}^{n}{\left|f_{i}-{\hat{f}}_{i}\right|}^{2}}$$
(21)$$MAPE=\frac{1}{n}\sum_{i=1}^{n}\left|\frac{{(f}_{i}-{\hat{f}}_{i})}{f_{i}}\right|\times 100\%$$

### 4.4. Results of GRU Prediction

REWS at different average wind speeds is automatically decomposed into several IMF components and one residual component (the number of IMFs depends on the constraint conditions of IMF [18]). The extracted IMFs from EMD at the average wind speed of 20 m/s are shown in Figure 6. In order to guarantee the feasibility of the experiment, the same GRU neural network is used for prediction under different average wind speeds via frequency grouping. 

The learning rate of the GRU neural network optimized using EO and particle swarm optimization (PSO) is shown in Figure 7. From Figure 7, the convergence rate of PSO is faster than EO, but there is premature convergence in PSO. Moreover, compared with PSO, EO has a better search capability, with about 0.02 smaller calculated fitness.

Figure 8 shows the GRU evaluation indicators of each group of the EMD–GRU scheme with lidar measuring information and calculated REWS as the combined input. From Figure 8, it is seen that the prediction result of GRU model is better under the low wind speed such as H1 and H2 than the high wind speed such as H5 and H6. GRU model has small prediction errors for the high-frequency group and residual but poor results for the low-frequency group, especially at high wind speeds. The reason is that the amplitude of the high-frequency component is much smaller than that of the low-frequency component, and the high-frequency proportion of the input data is very low.

#### 4.4.1. Results of Aggregation Optimization

Weights of aggregation optimization are shown in Figure 9a,b, which refers to the weights and the average weights of different groups under the six datasets, respectively. According to Figure 9, in different frequency groups, the high-frequency group has the largest average weight, and the low frequency group has the smallest average weight. Combined with the results of GRU prediction, it can be learned that the result of EO optimization is related to the effect of GRU prediction. In the optimization aggregation, the smaller the prediction error is, the higher the corresponding weight will be.

In order to demonstrate the advantages of employing EO, the EO algorithm and PSO algorithm are compared. As a typical optimization algorithm, PSO has wide applicability and reliability in various optimization problems, which is sufficient to verify the effectiveness of EO algorithm in parameter optimization. Table 3 shows the evaluation indicators before and after optimization. Under the same number of iterations, the optimization effect of EO is better than that of PSO. Under high wind speed, the optimization effect of EO is improved more significantly than that of PSO. For example, in the dataset H5, after performing EO optimization, the RMSE is decreased by 0.0592, which is about 0.03 lower than that of PSO optimization. 

#### 4.4.2. Prediction Results of EMD–GRU Schemes

Figure 10 shows the prediction results of different schemes under different average wind speeds. From Figure 10a–f, the prediction effect of Scheme 3 is the best, while that of Scheme 1 is the worst compared to the other two schemes. In Figure 10a, the variation of wind speed is gentle, and the prediction accuracy of the three schemes is higher compared to other different average wind speed. When the oscillation degree of wind speed curve amplitude increases, such as in Figure 10b,c,e,f, Scheme 1 is not suitable to predict. Scheme 3 has better stability than Scheme 2 according to Figure 10d.

The evaluation indicators of the three EMD–GRU schemes with different inputs are shown in Table 4. The prediction accuracy, stability, and effectiveness of Scheme 3 under different average wind speeds are better than those of the other two schemes. The prediction effect of Scheme 2 is close to Scheme 3 under low wind speed. The prediction effect of Scheme 1 is not accurate, especially under high wind speed.

Based on Figure 10 and Table 4, conclusions can be obtained: From the aspect of modeling accuracy, the average MAE of Scheme 3 is 0.2781, which represents the highest modeling accuracy, while that of Scheme 1 is 0.6629, representing the lowest modeling accuracy among the three schemes. The MAEs of Scheme 2 under H1–H6 are 0.2401, 0.3302, 0.3621, 0.4136, 0.3881, and 0.3271, respectively. Among these six datasets, the prediction accuracy of Scheme 2 performs best at an average wind speed of 10 m/s.From the aspect of modeling stability, when the lidar information is combined with the mechanism modeling, the prediction stability is obviously improved. The RMSE of Scheme 1 is approximately from 0.72 to 0.95 at six different average wind speeds. However, the RMSEs of Scheme 2 and Scheme 3 are both distributed within 0.51. When the modeling stability is improved, the prediction result will be less sensible to the change in wind speed.From the aspect of modeling effectiveness, Scheme 3 has a better fitting effect compared to Scheme 2, and that of Scheme 1 is the worst. The average value of MAPE in Scheme 3 is 0.0198 in the six datasets, followed by 0.0242 in Scheme 2. The average value of MAPE in Scheme 1 is about twice that in Scheme 3.

#### 4.4.3. Comparations with Other Models

The EMD–GRU prediction scheme with lidar information and the calculated REWS as the input is the best, so this scheme is selected for the experiment comparation with other models. In this comparation, the prediction performance of the EMD–GRU hybrid model, the GRU model, and the mechanism model are compared under the six datasets so as to illustrate the superiority of the proposed EMD–GRU hybrid model.

Figure 11 shows the actual curves and prediction curves of the three models under three average wind speeds of 12 m/s, 16 m/s, and 20 m/s. It can be learned that the prediction result of the mechanism model is the worst among all models. In Figure 11a, although mechanism modeling can reflect the change of wind speed, its volatility is larger than that of the other two models. In Figure 11b, the prediction effect of the three models is better than that of (a) and (c). In Figure 11c, although the actual wind speed varies greatly, the accurate prediction can still be achieved through a EMD–GRU hybrid model. 

Table 5 shows the comparison evaluations of REWS prediction using the three mentioned models under different average wind speeds. With an increase in average wind speed, the prediction error of mechanism model also increases. The GRU model has wonderful prediction performance under low wind speed. The EMD–GRU hybrid model further improves the prediction accuracy of the GRU data-driven model.

Specifically, three observations can be obtained:Compared to the traditional mechanism modeling, the proposed EMD–GRU model has significantly improved the prediction performance. For example, when the average wind speed is 12 m/s, the RMSE and MAE of the mechanism model are 0.6663 and 0.5432, respectively, while the RMSE and MAE of the EMD–GRU model are 0.3058 and 0.2525, respectively.Compared to the original GRU data-driven model, the predicted value of the EMD–GRU model is more consistent with the actual value of the REWS. From the average wind speed of 10 m/s to 20 m/s, the improvement rates of MAPE corresponding to the EMD–GRU model are 7.14%, 19.46%, 5.49%, 15.38%, 0.93%, and 8.81%, respectively.The EMD–GRU model has higher predictive stability than the other two models. For example, under the average wind speed of 10 m/s, compared to those of the other two models, the RMSE of the EMD–GRU model is decreased by 0.2629 and 0.0131, respectively.

The possible reasons for the above observations are as follows. Since the limited number of lidar measurement points and the limitation of the Taylor frozen turbulence hypothesis, the prediction error of mechanism modeling is large. Since the input information is quite nonlinear, the GRU data-driven method cannot avoid the influence of signal fluctuation. Differently from the two counterparts, the EMD–GRU scheme can effectively reduce signal volatility and predict error through frequency decomposition and classification, group prediction, and optimization aggregation of prediction components.

## 5. Conclusions

In this paper, a data-driven approach has been proposed to predict the REWS with lidar measurements. Three EMD–GRU schemes are proposed to improve the reliability of the REWS prediction. Accordingly, the main conclusions can be summarized as follows:➢Among three EMD–GRU schemes with different input, the prediction accuracy, stability, and effectiveness of Scheme 3 exhibit obvious superiority compared to those of the other two schemes.➢The EO and PSO algorithms could effectively optimize the prediction performance of EMD–GRU model, and the optimization effect of EO algorithm is better than that of PSO. The RMSE of the EMD–GRU model after EO optimization is reduced by 0.0592, which is about 0.03 lower than that of PSO.➢Compared to the traditional mechanism model and the single GRU model, the prediction performance of the proposed EMD–GRU model is significantly improved. Relative to the mechanism model, the EMD–GRU model demonstrates MAE improvements of 49.18%, 53.43%, 52.10%, 65.95%, 48.18%, and 60.33% across the six datasets.

Compared to some traditional models, the proposed EMD–GRU model, which includes data processing steps and neural network training processes, may require more computational resources and time to complete the prediction task. In real-time control applications, this method has some limitations and room for improvement. The future work focuses on using REWS prediction with lidar measurement to further optimize the control systems and achieve smarter and more sensitive control strategies to improve the performance and efficiency of wind turbines.

## Figures and Tables

**Figure 1 sensors-23-09379-f001:**
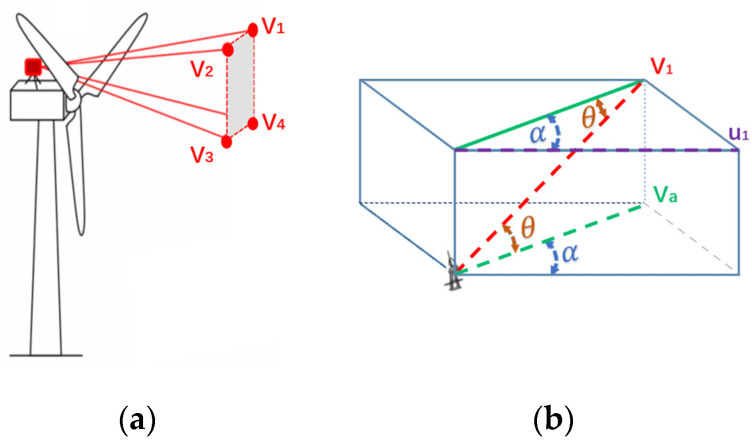
Diagram of measuring wind speed with laser beams: (**a**) the diagram with four wind speeds; (**b**) the diagram with $V_{1}$ only.

**Figure 2 sensors-23-09379-f002:**
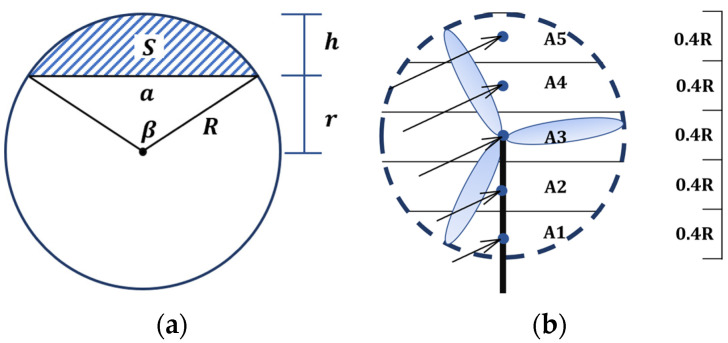
Diagram of REWS calculation: (**a**) Sector area; (**b**) REWS.

**Figure 3 sensors-23-09379-f003:**
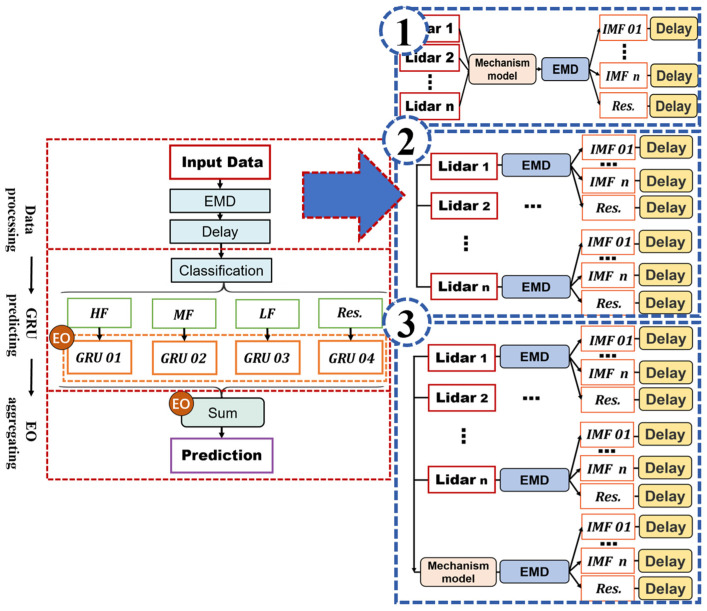
Basic structure of EMD–GRU model. The res., HF, MF, and LF respectively represent residual, high frequency, medium frequency, low frequency.

**Figure 4 sensors-23-09379-f004:**
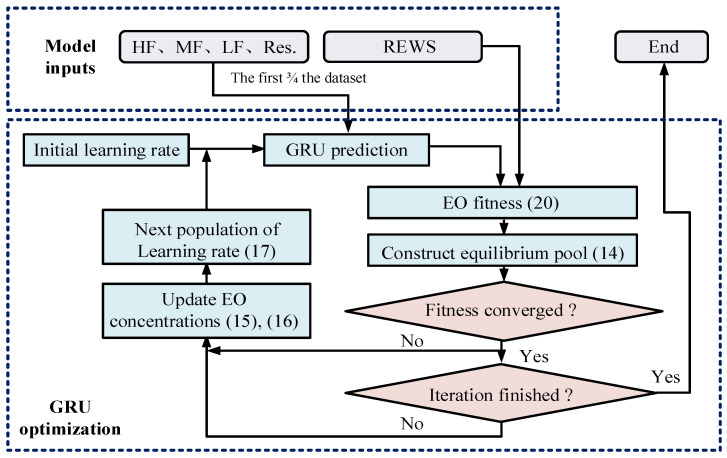
Optimization of GRU learning rate based on EO.

**Figure 5 sensors-23-09379-f005:**
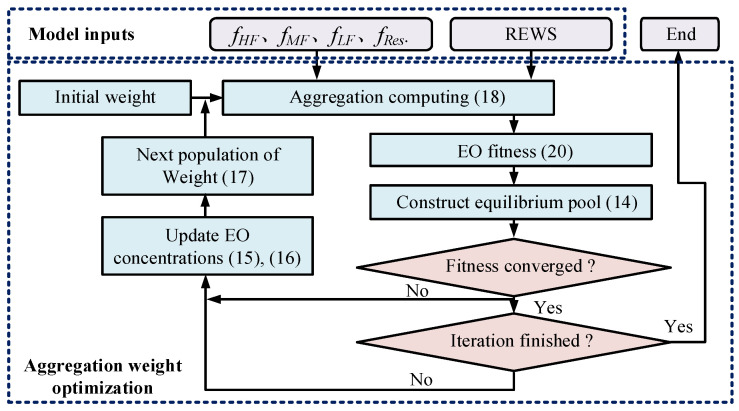
Optimization of aggregation weight based on EO.

**Figure 6 sensors-23-09379-f006:**
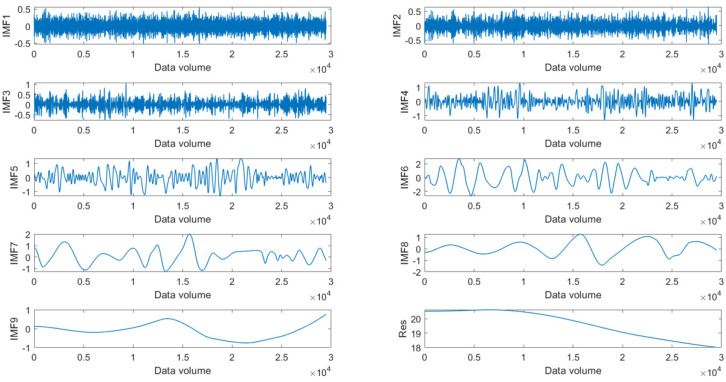
The extracted IMFs from EMD at the average wind speed of 20 m/s.

**Figure 7 sensors-23-09379-f007:**
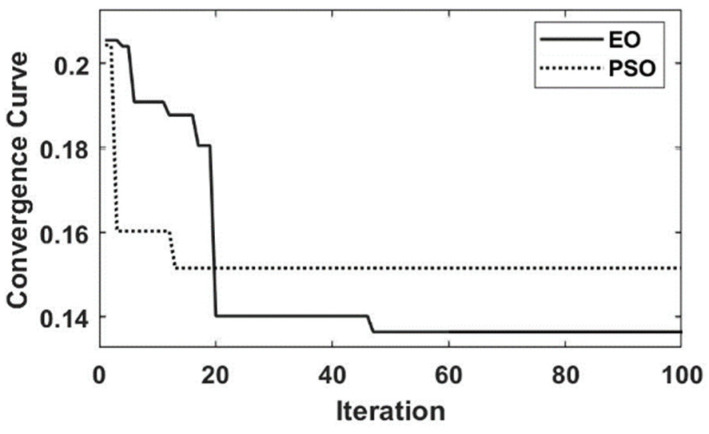
The optimized learning rate curves of GRU.

**Figure 8 sensors-23-09379-f008:**
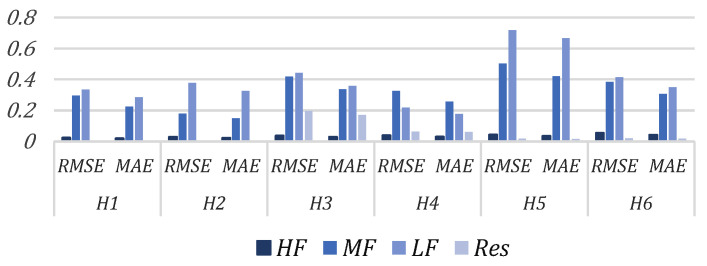
GRU prediction evaluation indicators of each component in Scheme 3.

**Figure 9 sensors-23-09379-f009:**
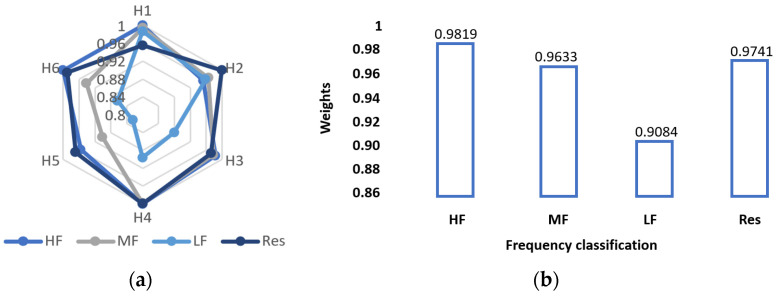
Weights of aggregation optimization in Scheme 3: (**a**) weights of each component under the six datasets; (**b**) average weights of different components.

**Figure 10 sensors-23-09379-f010:**
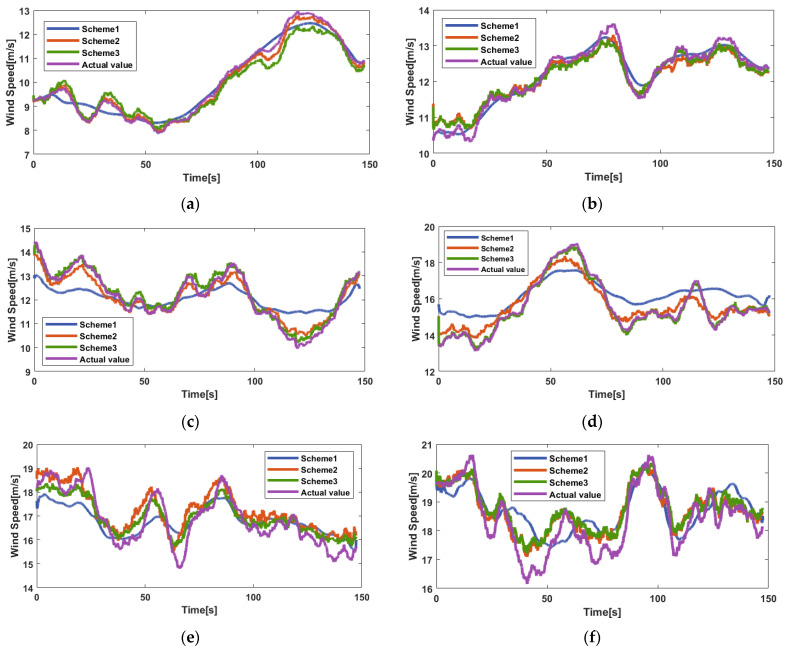
Prediction results of different schemes under different datasets: (**a**) 10 m/s; (**b**) 12 m/s; (**c**) 14 m/s; (**d**) 16 m/s; (**e**) 18 m/s; (**f**) 20 m/s.

**Figure 11 sensors-23-09379-f011:**
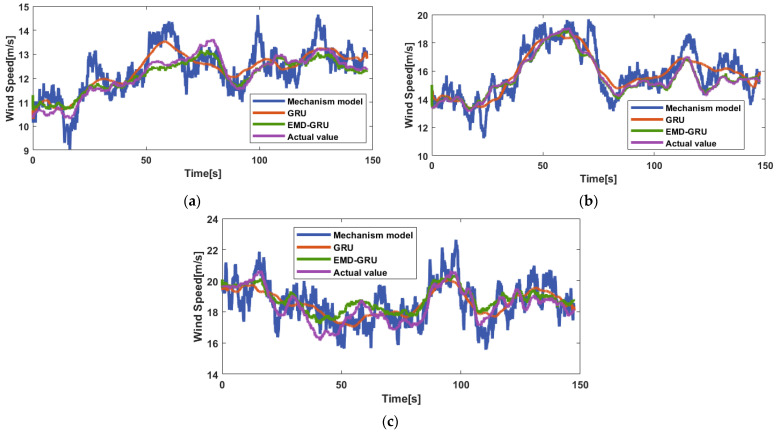
The actual curves and prediction curves of REWS under three different average wind speeds: (**a**) 12 m/s; (**b**) 16 m/s; (**c**) 20 m/s.

**Table 1 sensors-23-09379-t001:** Statistical characteristics of six wind speed datasets.

Datasets	Wind Speed	Samples	Mean	Max	Median	Min	Std.
H1	10	All	9.6115	12.9672	9.4057	7.8553	1.0714
		Training Set	9.4598	11.0997	9.3919	7.8626	0.7962
		Testing Set	10.0667	12.9672	9.5563	7.8553	1.5535
H2	12	All	12.0045	14.0607	11.9957	9.8206	0.8960
		Training Set	11.9482	14.0607	11.9000	9.8206	0.9044
		Testing Set	12.1734	13.6158	12.4482	10.3123	0.8483
H3	14	All	13.3294	17.3582	13.2469	9.9518	1.2814
		Training Set	13.7240	17.3582	13.5764	11.4107	1.1030
		Testing Set	12.1455	14.4256	12.1977	9.9518	1.0241
H4	16	All	15.8433	19.0663	15.8330	12.2807	1.5796
		Training Set	15.9308	19.0299	16.1401	12.2807	1.6052
		Testing Set	15.5808	19.0663	15.2350	13.1170	1.4693
H5	18	All	17.6392	23.5735	17.5249	13.9425	1.9126
		Training Set	17.8968	23.5735	18.1743	13.9425	2.0529
		Testing Set	16.8665	19.0372	16.7214	14.7905	1.0917
H6	20	All	19.5343	24.3731	19.4811	15.2719	1.7119
		Training Set	19.9127	24.3731	19.8991	15.2719	1.7167
		Testing Set	18.3988	20.6271	18.3912	16.1519	1.0780

**Table 2 sensors-23-09379-t002:** Parameter setting of the proposed model.

Model	Parameter Name	Parameter Value
GRU	Hidden Units	230
	Learning Rate Drop Period	4
	Epoch	60
EO_1_	*n*	20
	$C_{min}$	0.001
	$C_{max}$	0.01
	Max_iter	100
EO_2_	*n*	20
	$C_{min}$	0
	$C_{max}$	5
	Max_iter	50

**Table 3 sensors-23-09379-t003:** Comparison results of EO and PSO.

DATA	Not Optimized	EO	PSO
H1	0.2804	0.2803	0.2804
H2	0.3889	0.3058	0.3875
H3	0.3547	0.3443	0.3501
H4	0.3139	0.2866	0.3078
H5	0.4865	0.4273	0.4837
H6	0.4319	0.4032	0.4276

**Table 4 sensors-23-09379-t004:** Comparison results of three EMD–GRU schemes for six datasets.

Data	Model	RMSE	MAE	MAPE
H1	Scheme 1	0.7356	0.6605	0.0586
	Scheme 2	0.2855	0.2401	0.0233
	Scheme 3	0.2803	0.2212	0.0208
H2	Scheme 1	0.7855	0.6720	0.0540
	Scheme 2	0.3898	0.3302	0.0266
	Scheme 3	0.3058	0.2525	0.0207
H3	Scheme 1	0.7371	0.6065	0.0513
	Scheme 2	0.3981	0.3621	0.0272
	Scheme 3	0.3443	0.2898	0.0241
H4	Scheme 1	0.7828	0.6555	0.0420
	Scheme 2	0.5026	0.4136	0.0270
	Scheme 3	0.2866	0.2236	0.0143
H5	Scheme 1	0.7258	0.5942	0.0354
	Scheme 2	0.4726	0.3881	0.0229
	Scheme 3	0.4273	0.3606	0.0214
H6	Scheme 1	0.9448	0.7887	0.0434
	Scheme 2	0.4037	0.3271	0.0179
	Scheme 3	0.4032	0.3211	0.0176

**Table 5 sensors-23-09379-t005:** Comparison results of different prediction schemes for six datasets.

Data	Model	RMSE	MAE	MAPE
H1	Mechanism	0.5432	0.4353	0.0452
	GRU	0.2934	0.2265	0.0224
	EMD–GRU	0.2803	0.2212	0.0208
H2	Mechanism	0.6663	0.5423	0.0454
	GRU	0.3891	0.3229	0.0257
	EMD–GRU	0.3058	0.2525	0.0207
H3	Mechanism	0.7570	0.6051	0.0461
	GRU	0.3686	0.3027	0.0255
	EMD–GRU	0.3443	0.2898	0.0241
H4	Mechanism	0.8315	0.6567	0.0417
	GRU	0.3317	0.2609	0.0169
	EMD–GRU	0.2866	0.2236	0.0143
H5	Mechanism	0.8741	0.6959	0.0394
	GRU	0.4871	0.3895	0.0216
	EMD–GRU	0.4273	0.3606	0.0214
H6	Mechanism	0.9972	0.8095	0.0419
	GRU	0.4393	0.3359	0.0193
	EMD–GRU	0.4032	0.3211	0.0176

## Data Availability

The datasets used and/or analyzed during the current study are available from the corresponding author on reasonable request.

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
