# Peer review of "From Lidar Measurement to Rotor Effective Wind Speed Prediction: Empirical Mode Decomposition and Gated Recurrent Unit Solution"

_sensors, 2023, doi:10.3390/s23239379_

Round 1

Reviewer 1 Report

Comments and Suggestions for Authors

The paper deals with the rotor effective wind speed prediction for wind turbine applications, by proposing a novel approach based on input Lidar data and a framework of a  combined Empirical Mode Decomposition and Gated Recurrent Unit neural network. The proposed approach and the obtained results seem reliable and accurate, showing better performances compared with PSO algorithm. The paper results are extensively discussed and well justified, in a clear and simple approach; however, the paper should be carefully revised for the sake of accuracy and clarity.

The following issues are recommended to improve the paper:

1.     Title: if convenient, please avoid the use of acronyms in the paper title.

2.     Abstract: the abstract should be devoted, among others (objective, methods, results, conclusions), to briefly present the context or background information for your research (and their criticism) and to highlight the paper novelty. Please address here also quantitative results, i.e. at least the precision of results vs. other methods.

3.     “the large inertia of rotor rotation” Proposal to reformulate, as inertia is “a property of matter by which it continues in its existing state of rest or uniform motion in a straight line, unless that state is changed by an external force.” (is not a property of motion, i.e. “rotation”!)

4.     Recommendation to include a Nomenclature section for the used acronyms and symbols.

5.     Few typing mistakes should be fixed. E.g., use “Eq. (x)” instead of simple “(x)”, use space between a value and its measurement unit, renumber “4.4.2. Comparations with other models”, as there is also “4.4.2. Prediction results of EMD-GRU schemes”, etc.

6.     “In order to demonstrate the advantages of employing EO, the EO algorithm and PSO algorithm are compared.” Justify the selection of PSO and why other algorithms have been neglected.

7.     “Compared with other average wind speeds, the prediction accuracy of scheme 2 is better at the average wind speed of 10 m/s.” This conclusion is not supported by the data in Table IV, please clarify it!

8.     Conclusions: extend this section with the most relevant results of the research, emphasize the limits of the proposed approach and highlight future research directions.

Author Response

Many thanks for your time and efforts in handling our paper and your encouragements. Following your suggestion, we have improved our paper. Please check the revised paper for the details.

  1. Title: if convenient, please avoid the use of acronyms in the paper title.
  2. Abstract: the abstract should be devoted, among others (objective, methods, results, conclusions), to briefly present the context or background information for your research (and their criticism) and to highlight the paper novelty. Please address here also quantitative results, i.e. at least the precision of results vs. other methods.
  3. “the large inertia of rotor rotation” Proposal to reformulate, as inertia is “a property of matter by which it continues in its existing state of rest or uniform motion in a straight line, unless that state is changed by an external force.” (is not a property of motion, i.e. “rotation”!)
  4. Recommendation to include a Nomenclature section for the used acronyms and symbols.
  5. Few typing mistakes should be fixed. E.g., use “Eq. (x)” instead of simple “(x)”, use space between a value and its measurement unit, renumber “4.4.2. Comparations with other models”, as there is also “4.4.2. Prediction results of EMD-GRU schemes”, etc.
  6. “In order to demonstrate the advantages of employing EO, the EO algorithm and PSO algorithm are compared.” Justify the selection of PSO and why other algorithms have been neglected.
  7. “Compared with other average wind speeds, the prediction accuracy of scheme 2 is better at the average wind speed of 10 m/s.” This conclusion is not supported by the data in Table IV, please clarify it!
  8. Conclusions: extend this section with the most relevant results of the research, emphasize the limits of the proposed approach and highlight future research directions.

Reviewer 2 Report

Comments and Suggestions for Authors

- Need to avoid short forms from the title

- need to rewrite abstract to show the novelty of the work and also include few statistics related to the performance of the proposed approach

-what is the reason behind to select the EMD for the data processing? why not other. compare EMD with wavelet

-need to redraw figure 8 to make readable and also mention the axis in fig 8(b)

- need to rewrite conclusion to show the novelty of the work and also include few statistics related to the performance of the proposed approach

- kindly include the waveform of the extracted IMFs from EMD

- compare the results with and without using EMD. show the results in tabular form

Comments on the Quality of English Language

Good work

Author Response

Many thanks for your time and efforts in handling our paper and your encouragements. Following your suggestion, we have improved our paper. Please check the revised paper for the details.

- Need to avoid short forms from the title

- need to rewrite abstract to show the novelty of the work and also include few statistics related to the performance of the proposed approach

-what is the reason behind to select the EMD for the data processing? why not other. compare EMD with wavelet

-need to redraw figure 8 to make readable and also mention the axis in fig 8(b)

- need to rewrite conclusion to show the novelty of the work and also include few statistics related to the performance of the proposed approach

- kindly include the waveform of the extracted IMFs from EMD

- compare the results with and without using EMD. show the results in tabular form

Reviewer 3 Report

Comments and Suggestions for Authors

In this paper, a data-driven approach has been proposed to predict REWS with 443 Lidar measurements.

The work is presented on a frontier topic for new technologies and measurement methods for wind speed as a prediction for inputs in control systems, which is relevant for the area of ​​wind energy, however the paper needs improvement. . in some aspects:

Figure 4, 5, they must improve their quality

The figures of the predictions as a result for different data sets should include different colors for a better understanding of the trajectories, while black and white are confused despite using different patterns.

The conclusions section is poor. Future work should be talked about and perhaps the perspective regarding control systems. Furthermore, in the discussions or conclusions it would be necessary to talk about the processes and times for this type of models in comparison to others. Do methods for wind speed, cost, processing and complexity compete against current wind profile input systems in current turbines?

Author Response

Many thanks for your time and efforts in handling our paper and your encouragements. Following your suggestion, we have improved our paper. Please check the revised paper for the details.

In this paper, a data-driven approach has been proposed to predict REWS with 443 Lidar measurements.

The work is presented on a frontier topic for new technologies and measurement methods for wind speed as a prediction for inputs in control systems, which is relevant for the area of ​​wind energy, however the paper needs improvement. . in some aspects:

Figure 4, 5, they must improve their quality

The figures of the predictions as a result for different data sets should include different colors for a better understanding of the trajectories, while black and white are confused despite using different patterns.

The conclusions section is poor. Future work should be talked about and perhaps the perspective regarding control systems. Furthermore, in the discussions or conclusions it would be necessary to talk about the processes and times for this type of models in comparison to others. Do methods for wind speed, cost, processing and complexity compete against current wind profile input systems in current turbines?

Round 2

Reviewer 3 Report

Comments and Suggestions for Authors

The work improved in quality and all recommendations and changes were made appropriately, therefore, my review indicates that it is possible to publish the work.